# Biochemical Functions and Clinical Characterizations of the Sirtuins in Diabetes-Induced Retinal Pathologies

**DOI:** 10.3390/ijms23074048

**Published:** 2022-04-06

**Authors:** Samanta Taurone, Chiara De Ponte, Dante Rotili, Elena De Santis, Antonello Mai, Francesco Fiorentino, Susanna Scarpa, Marco Artico, Alessandra Micera

**Affiliations:** 1IRCCS—Fondazione Bietti, via Livenza 3, 00198 Rome, Italy; alessandra.micera@fondazionebietti.it; 2Department of Sensory Organs, Sapienza University of Rome, Piazzale Aldo Moro 5, 00185 Rome, Italy; chiara.deponte@uniroma1.it (C.D.P.); marco.artico@uniroma1.it (M.A.); 3Department of Drug Chemistry and Technologies, Sapienza University of Rome, Piazzale Aldo Moro 5, 00185 Rome, Italy; dante.rotili@uniroma1.it (D.R.); antonello.mai@uniroma1.it (A.M.); f.fiorentino@uniroma1.it (F.F.); 4Department of Anatomical, Histological, Forensic Medicine and Orthopedic Sciences, Sapienza University of Rome, Piazzale Aldo Moro 5, 00185 Rome, Italy; elena.desantis@uniroma1.it; 5Experimental Medicine Department, Sapienza University of Rome, Viale Regina Elena 324, 00161 Rome, Italy; susanna.scarpa@uniroma1.it

**Keywords:** antioxidants, anti-VEGF, diabetes mellitus, diabetic retinopathy, free radicals, neuroinflammation, oxidative stress, sirtuins

## Abstract

Diabetic retinopathy (DR) is undoubtedly one of the most prominent causes of blindness worldwide. This pathology is the most frequent microvascular complication arising from diabetes, and its incidence is increasing at a constant pace. To date, the insurgence of DR is thought to be the consequence of the intricate complex of relations connecting inflammation, the generation of free oxygen species, and the consequent oxidative stress determined by protracted hyperglycemia. The sirtuin (SIRT) family comprises 7 histone and non-histone protein deacetylases and mono (ADP-ribosyl) transferases regulating different processes, including metabolism, senescence, DNA maintenance, and cell cycle regulation. These enzymes are involved in the development of various diseases such as neurodegeneration, cardiovascular pathologies, metabolic disorders, and cancer. SIRT1, 3, 5, and 6 are key enzymes in DR since they modulate glucose metabolism, insulin sensitivity, and inflammation. Currently, indirect and direct activators of SIRTs (such as antagomir, glycyrrhizin, and resveratrol) are being developed to modulate the inflammation response arising during DR. In this review, we aim to illustrate the most important inflammatory and metabolic pathways connecting SIRT activity to DR, and to describe the most relevant SIRT activators that might be proposed as new therapeutics to treat DR.

## 1. Introduction

### 1.1. Diabetic Retinopathy (DR)

Diabetic eye disease (DED) indicates a variety of eye conditions that affect people with diabetes mellitus (DM), including diabetic retinopathy (DR), diabetic macular edema (DME), cataracts, and glaucoma. DR refers to the pathological and functional changes observed in the retinal vascular system due to chronic hyperglycemia and is recognized as the most frequent microvascular complication of DM. It affects approximately one third of type 1 and type 2 diabetic patients.

Diabetic retinopathy represents the main cause of blindness in subjects under the age of 50. It is expected that DR incidence will increase in the next few decades due to the increase of diabetes in the world population, affecting approximately 191 million people worldwide by 2030 [1,2,3].

Diabetic retinopathy is classified into two forms, one early and less severe (non-proliferating) (NPDR) and one advanced (proliferating) (PDR). Non-proliferative diabetic retinopathy is divided into mild, moderate, or advanced; if not recognized and treated promptly, evolves towards the highly disabling proliferating form. In NPDR retinopathy, hyperglycemia damages the structure of blood vessels predisposing to the formation of microaneurysms, microhemorrhages, and anomalies of the retinal vascular caliber [1,2,3,4]. These anomalies can lead to the passage, through the damaged walls of the vessels themselves, of some components of the blood, or to reduced perfusion of the retinal tissue up to a complete ischemia, which initially manifests itself with the presence of cottony exudates. The identification of advanced non-proliferative retinopathy is important as we know that it evolves, if left untreated, into proliferative diabetic retinopathy in 40% of cases within 12 months [5]. The occlusion of retinal capillaries and the consequent formation of ischemic retinal areas represent the stimulus for the formation of retinal new vessels, which characterize the proliferating form [1,2,3].

The newly formed blood vessels, not possessing an adequate structure, can easily rupture, with the risk of developing pre-retinal hemorrhages and secondary retinal detachments. In the early stages, diabetic retinopathy is generally asymptomatic. Furthermore, the lack of symptoms is not a reliable indication of the absence of diabetic retinal microangiopathy, since the reduction of vision, which the patient notices, appears only when the macular region (central part of the retina) is affected. An early diagnosis and an exact classification of diabetic retinopathy are therefore essential elements to prevent the occurrence of ocular damage, which can lead in extreme cases to blindness [1,2,3,5]. The most effective weapons to reduce the frequency of onset or aggravation of the disease certainly remain, as mentioned above, accurate prevention and rigorous metabolic compensation.

### 1.2. Neurodegeneration in DR

To date, the precise mechanisms by which neuroinflammation and vascular inflammation occur in DR are not known.

In patients with retinal disease, degeneration and loss of function of retinal neurons, especially retinal ganglion cells (RGC), are highlighted [6].

Some studies suggest that neuronal damage develops before vascular damage, attributing a neuropathic origin to the disease [6,7], highlighting the crosstalk between the neuronal, glial, and vascular cells that make up the NVU (Neurovascular unit) retinal [8], some researchers hypothesize that damage to non-clinically diagnosable endothelial cells may induce activation of microglial cells [9]. From the data in the literature, diabetic retinopathy could be caused by the activation of the inflammatory response both at the vascular and at the neuronal level.

Several research groups support the idea that, at the retinal level, the neurodegenerative process can be triggered by vascular inflammation, and the same process occurs in neurodegenerative diseases at the brain level. Since both glial cells and vascular endothelial cells are closely associated, Barber et al. suggest that the reactivity of the former (glial cells) is a direct consequence of the infiltration of glucose and inflammatory agents into the nerve parenchyma and that, in turn, the increase in vascular permeability is supported by the release of some glial factors with the consequent loss integrity of the blood-retinal barrier [9]. In conditions of hyperglycemia, microglia is activated and secretes cytokines and other pro-inflammatory molecules used for phagocytosis and for the destruction of damaged cells, as well as for the initiation of repair processes that lead to the formation of glial scars. If the microglia remains in an activated state, however, the cytokines can damage neighboring cells, particularly neuronal cells, leading to the appearance of other retinal diseases, such as retinal degeneration and glaucoma [10]. In accordance with this thesis, numerous histopathological studies conducted on animals and in humans have highlighted the activation of microglial cells, as well as the presence of various inflammatory molecules secreted by microglia [11,12,13]. This condition induces the secretion of tumor necrosis factor-α (TNF-α) and the localized secretion of other proinflammatory cytokines, growth factors and bioactive molecules that play important roles in the onset and progression of diabetic retinopathy [14,15]. The release of cytokines and pro-inflammatory molecules such as TNF-α, interleukin-1β (IL-1β), nitric oxide (NO) and vascular endothelial growth factor (VEGF) causes the spread of the inflammatory process through the entire retina, aggravating the increase in vascular permeability and neuronal damage and thus creating a vicious circle [12]. Müller cells and astrocytes under stress become active and produce proinflammatory cytokines and growth factors to restore tissue homeostasis in chronic diseases such as diabetic retinopathy. The persistent inflammatory response then leads to death or cell damage [16].

Liou et al. have recently confirmed that microglia activation occurs early in the diabetic pathology, producing a wide range of proinflammatory cytokines such as IL-1β, interleukin-3 (IL-3), interleukin-6 (IL-6), TNF-α and other inflammatory mediators such as reactive oxygen species (ROS, glutamate, (VEGF), metalloproteinase and NO. These mediators induce the expression of adhesion molecules (I-CAM and V-CAM), cell apoptosis, leukocyte infiltration and weakening of the blood–retinal barrier [17].

## 2. Microvascular Alteration and Inflammation in DR

DR usually begins with the appearance of retinal microaneurysms and the presence of spot hemorrhages [2]. Subsequently, the disease evolves into the most severe form of proliferative DR. During this phase, neovascular changes occur with consequent deposition of fibrotic tissue, retinal detachments, vitreous hemorrhages, and diabetic macular edema [2,18,19]. The retinal tissues of diabetic patients show increased thickening of the basement membrane (BM) of capillaries. Normally the retinal capillaries are made up of continuous internal layers of endothelial cells enclosed by the irregular processes of intramural pericytes. The BM consists of two layers: a thin subendothelial internal basal membrane (IBM), inserted between the endothelial cells and the pericytes, and an external basal membrane (EBM) positioned between the pericytes and the Müller glial cells. Under pathological conditions, a significant thickening of the EBM is observed [2,20,21,22]. The thickening of the BM mainly involves the diabetic capillaries located in the layer of nerve fibers near the internal limiting membrane. Furthermore, retinas of diabetic subjects are characterized by an alteration of the microvascular subcomponents of the extracellular matrix (ECM) along with a noticeable loss of pericytes [2,3,20,21,22]. During the preclinical phase of diabetic retinopathy (PCDR), the increase in permeability of the BRB that determines the accumulation of numerous pro-angiogenic and inflammatory cytokines [1]. Overall, the main factors contributing to the initiation and advancement of DR are inflammation, oxidative stress, and persistent hyperglycemia, which leads to increased levels of advanced glycation end products (AGEs) through non-enzymatic glycation [23].

AGEs are particularly dangerous since the modification of cellular matrix proteins, soluble proteins, and DNA alter their physicochemical properties and, consequently, their cellular functions. AGE-modified proteins may stimulate apoptosis due to the interaction with specific factors name receptors for advanced glycation end products (RAGE). This pathway causes an upsurge of ROS, such as nitric oxide (NO), hydrogen peroxide (H_2_O_2_), and the superoxide anion (O_2_^−^) [24]. AGEs abundance leads to an increased activation of hexosamine and polyol pathways with consequent release of angiopoietin-2, stimulation of the renin–angiotensin system, and eventually leading to the formation of new blood vessels. Furthermore, protein kinase C (PKC) is activated and, in turn, stimulates endothelial proliferation and neovascularization. These events determine increased vascular permeability and aberrant blood flow, thereby further increasing oxidative stress, and finally stimulating apoptosis [25].

Oxidative stress also leads to the activation of nuclear factor-κB (NF-κB), a pro-apoptotic factor that is also activated in retinal pericytes because of high glucose levels. Stimulation of NF-κB pathway in retinal endothelial cells supports high expression of proinflammatory cytokines such as interleukin-8 (IL-8), IL-6, TNF-α and many pro-apoptotic modulators. The action of TNF-α changes the properties of the retinal endothelial barrier [1,2,3]. In particular, the expression of the tight junction proteins claudin-5 and zonula occludens-1 is reduced, and their distribution is also altered. The outcome of this process consists of augmented vascular permeability in the retina [26]. As mentioned above, inflammation is associated with high levels of cytokines and proinflammatory mediators, along with major leukocytes adhesion. The consequent enhancement of vascular permeability and the rise in ROS production ultimately lead to cellular damage and apoptosis. High IL-1β concentrations are detected in retinas of diabetic animal models along with an enhanced caspase-1 activity. IL-1β is among the proinflammatory cytokines able to activate NF-κB, thereby stimulating the transcription of other cytokines and proinflammatory factors such as intercellular adhesion molecule-1 (ICAM-1) and inducible nitric oxide synthase (iNOS) [1,2,3,27].

## 3. Sirtuins Overview

Sirtuins (SIRTs), also named class III histone deacetylases (HDACs), utilize nicotinamide adenine dinucleotide (NAD^+^) as a cofactor for their catalytic functions that consist in both histone and non-histone proteins’ deacylation and mono-ADP-ribosylation. This enzyme family comprise 7 isoforms whose activity is mainly connected to the modulation of DNA repair, cell cycle, metabolism, and aging. Given their involvement in a wide range of processes, the activity of SIRTs has been connected to various disorders, including neurodegenerative diseases, cardiovascular disorders, metabolic pathologies, and cancer [28].

SIRT1, 3, 5, and 6 are central regulators of DR as they control the sensitivity to insulin, along with glycolysis, gluconeogenesis, and the initiation of the inflammatory process [29].

SIRTs are 7 protein deacylases and mono-ADP-ribosylases with high homology to the *Saccharomyces cerevisiae* Silent Information Regulator (SIR)-2 that possess different distribution patterns inside eukaryotic cells. These enzymes are involved in a wide range of cellular processes, including the modulation of cellular plasticity and the regulation of adaptation mechanisms in several stress-related pathways. SIRT1 and SIRT2 are present in both the nucleus and cytoplasm; they shuttle between nucleus and cytosol based on the cell cycle and tissue development phase. Differently, SIRT3-5 are mainly mitochondrial enzymes, while SIRT6 and 7are mainly present in nucleus [30,31]

SIRTs possess a catalytic core, which is approximately 275 residues long, while differs in the length of *C*- and *N*-terminal domains, that are isoform-specific and control their subcellular localization. Interestingly, neither SIRT4 nor SIRT5 possess a *C*-terminal domain, but they bear a small *N*-terminal sequence which acts as a mitochondrial tag [32].

The shared catalytic core contains the substrate binding site, along with the region accommodating the NAD^+^ cofactor. Although SIRTs were initially described as histone deacetylases, subsequent studies reported further activities such as the removal of succinyl, glutaryl, malonyl, and fatty acyl (e.g., myristoyl, palmitoyl) groups from both histone and non-histone proteins. SIRT4 and SIRT6 also show a mono-ADP-ribosyl transferase activity. The *C*- and *N*-terminal domains influence SIRTs cellular localization, substrate preference, binding to modulators, and interactions with other protein partners [33].

Although the catalytic mechanism is the same for all SIRTs, there are isoform-specific substrate preferences given the presence of little differences in the substrate binding site, which determine the binding affinity for different acylated substrates [34].

SIRT1, the first and best characterized human SIRT, is highly abundant in embryonic tissues, neurons, and in the nucleus of endothelial cells. SIRT1 activity regulates the maintenance of the euchromatin state and chromatin stability. In line with this, it regulates critical cellular processes, including DNA damage repair, inflammation, response to oxidative stress, senescence, and angiogenesis. Numerous reports indicate that SIRT1 is implicated in cancer, metabolic disorders, neurodegenerative diseases, and cardiovascular pathologies. SIRT1 has a pivotal function in different metabolic processes such as fatty acids β-oxidation, lipogenesis, and gluconeogenesis [35,36].

Indeed, studies conducted on murine models suggested that overexpression of SIRT1 is correlated with a decrease of adiposity, cholesterol, and insulin levels. Consequently, SIRT1-overexpressing mice presented improved clinical situation in terms of insulin resistance, glucose tolerance, and obesity. Moreover, SIRT1 inhibits the inflammation process through the deacetylation of important proteins such as NF-κB, AP-1, and STAT3 [37,38].

In line with this, SIRT1 expression maybe induced by oxidative stress. The rise of SIRT1 levels leads to an increase of antioxidant enzymes concentration, thereby protecting neurons from axonal degeneration. At the hepatic level, SIRT1 activity acts as a protective factor in the context of glucose intolerance and non-alcoholic fatty liver disease (NAFLD).

Remarkably, SIRT1 dysregulation has been linked to numerous ocular pathologies, including cataracts, glaucoma, age-related macular degeneration, and DR. Indeed, SIRT1-deficient mice are characterized by increased apoptosis of retinal cells as a consequence of p53 hyperacetylation and develop eye defects during embryogenesis.

SIRT2 is present in different tissues and organs such as adipose tissue, liver, prostate, kidneys, and, most importantly, in the central nervous system (CNS). One of the main targets of SIRT2 is α-tubulin, whose deacetylation contributes to the modulation of microtubules’ stability in neurons. SIRT2 has been shown to possess a central position in the onset and progression of neurodegenerative diseases. In line with this, inhibition SIRT2 has been indicated to protect neurons from α-synuclein-induced toxicity, a risk factor of familial Parkinson’s disease [39].

SIRT2 also regulates cell cycle and apoptosis through deacetylation of deacetylate histone H4 lysine 16 (H4K16) and p53 [40], and controls the expression of NF-κB-dependent genes through deacetylation [41] of its subunit p65. Similar to other SIRT isoforms, SIRT2 has a controversial role in cancer, since it plays either a tumor suppressor or promoter role depending on the physiopathological context [42].

Moreover, SIRT2 regulates cellular energetic state by modulating the action of insulin and the lipogenesis, hence it is regarded as a promising target in obesity, liver steatosis, and diabetes.

SIRT3 is the first reported SIRT to act in mitochondria where it contributes to the regulation of energy metabolism. Therefore, SIRT3 is mainly expressed in tissues characterized by a high metabolic rate such as muscles, heart, liver, brain, and brown adipose tissue. Depending on the cellular status, SIRT3 exists in both a long-chain and a short-chain form. The long-chain one is mainly positioned in the nucleus, while the short-chain one is obtained under cellular stress during which SIRT3, thanks to a *N*-terminal mitochondrial targeting sequence, is translocated to mitochondria where the mitochondrial matrix processing peptidase (MPP) cleaves the first 101 residues of SIRT3 [43].

Through its activity, SIRT3 controls fatty acid oxidation, oxidative phosphorylation, ketogenesis, and amino acid catabolism. Although mice in which SIRT3 gene is knocked out do not display phenotypic variations, they present altered insulin signaling pathway, along with decreased glucose tolerance, dyslipidemia, liver steatosis, and obesity. 

Under caloric restriction conditions, SIRT3 plays a protective role from oxidative stress. Indeed, through its deacetylase activity, it activates the mitochondrial isocitrate dehydrogenase 2 (IDH2), thereby increasing NADPH levels and augmenting the proportion of reduced-to-oxidized glutathione in mitochondria. Moreover, SIRT3-mediated deacetylation determines the activation of SOD and of the glutathione system which can prevent ROS accumulation.

SIRT4 was initially indicated to possess mitochondrial mono ADP-ribosyl-transferase, lipoamidase, and deacetylase activities, although it has been recently demonstrated that it possesses a significant broad-spectrum deacylase activity [44,45].

Recent studies point towards a possible role of SIRT4 as a regulator of cellular metabolism. In line with this hypothesis, SIRT4 has been shown to inhibit fat oxidation and to reduce insulin secretion in response to amino acids in pancreatic β-cells. SIRT4 may also promote the development of NAFLD, while SIRT3 and 5 play a protective role against it. Notably, SIRT4 lysine deacetylase activity is involved in the regulation of leucine metabolism, whose dysregulation causes elevated secretion of basal and stimulated insulin which in turn may lead to glucose intolerance and diabetes. In addition, SIRT4 impairs glutamine catabolism and regulates genome stability in normal and cancer cells in response to DNA damage [46].

SIRT4 has a dichotomous role in carcinogenesis since it acts both as a tumor suppressor or promoter, depending on the tissue context.

SIRT5 is among the most recent SIRTs to be characterized. Although it is mainly present in the mitochondria, it is also present in the cytosol. SIRT5 is mainly expressed in brain, heart, liver, kidney, and skeleton muscles. SIRT5 possesses a weak deacetylase activity while its actions are mainly mediated by the desuccinylation, demalonylation, and deglutarylation of the lysine residues of various protein substrates that play crucial roles in ROS detoxification and metabolism such as glycolysis, pentose phosphate pathway, ketone body formation, fatty acid β-oxidation, glutamine metabolism, and ammonia detoxification [47,48].

Proteomics analysis suggested that protein desuccinylation is connected to tricarboxylic acid cycle, amino acid metabolism, and fatty acid oxidation in heart and liver. Hepatic gluconeogenesis and glycolysis are also modulated by SIRT5 demalonylase activity. The desuccinylation activity promotes cellular resistance to oxidative stress through the inhibition of cellular respiration and through the activation of crucial enzymes such as SOD1.

The alteration of SIRT5 activity is also connected to the onset of neurodegenerative disorders, and, similar to other SIRTs, it has a dual role in cancer, acting both as tumor promoter and suppressor depending on the biological context.

SIRT6 is a nuclear deacylase and NAD^+^-dependent mono ADP-ribosyl transferase. Through its activity, it regulates DNA repair and genomic stability, but also metabolism, aging, inflammation, and immunity [49,50].

For this reason, it has been linked to neurodegenerative diseases and cancer, in which it shows a double-faced role, depending on the context [51].

Recent evidence indicates that the expression of SIRT6 is protective towards obesity and metabolic disorders and is linked to the suppression of hepatic glycolysis and gluconeogenesis. In line with this, SIRT6 knockout mice tend to develop glucose intolerance, liver steatosis, insulin resistance, and inflammation. SIRT6 also regulates TNF-α secretion through its demyristoylase activity, thereby affecting the inflammation processes [50] with its activity. Likewise, expression of SIRT6 in mice causes decreased inflammation in adipose tissue through the polarization of macrophages mediated by IL-4.

SIRT7 is predominantly located in the nucleolus and possesses deacetylase and desuccinylase activities towards specific protein substrates [52,53]. Its expression is prominent in testis, spleen, and liver. SIRT7 knockout mice are characterized by accumulation of DNA damage, early embryonic lethality, and reduced lifespan. Furthermore, low expression of SIRT7 is associated with hepatic steatosis and cardiac hypertrophy. SIRT7 may also be considered as an oncogene, given its overexpression in different cancer cell lines, along with its role in the maintenance of the cancerous phenotype [54].

Given the above-described evidence, SIRTs play a fundamental role in a diverse subset of biological settings and have been proposed as an attractive target for different therapeutic indications.

## 4. Sirtuins and Diabetic Retinopathy

### 4.1. SIRT1

Although the implication of SIRT1 activity in DR has been widely examined, the intricate molecular mechanisms underlying the development of this disease have not been completely clarified. Nonetheless, considering recent studies, there is a well-established connection between the expression of SIRT1 and the progress of DR and PDR [55].

Moreover, different reports point towards a connection between several microRNAs (miRNAs) and the onset of DM, DR, and other metabolic diseases. MiRNAs are short, single-stranded RNA molecules that negatively modulate the expression of specific genes. They act via base-pairing with target mRNAs, thereby inducing their degradation or inhibiting their translation [56].

For instance, Mortuza et al. [57] showed that the expression of SIRT1 in DR is controlled at a post-transcriptional level by miRNA-195. The authors of this study initially incubated human retinal endothelial cells (HRECs) with high glucose concentration which led to an upsurge in miRNA-195 concentration, along with a decrease in SIRT1 concentration. Notably, intravitreal injections of the anti-miRNA (antagomir) of miRNA-195 normalized the retinal levels of both miRNA-195 and SIRT1. Through the modulation of SIRT1 expression, miRNA-195 determines a rise in the production of fibronectin (FN), along with increased oxidative stress and vascular permeability, all of them being characteristic features of diabetic damage [57]. Similarly, in RECs, hyperglycemia determines SIRT1 down regulation followed by a decrease of mitochondrial antioxidant enzymes levels through pathways controlled by p300 and Fork head box protein O1 (FOXO1). In HRECs, SIRT1 expression is also controlled by miRNA-23b-3p [58]. In DR, miRNA-23b-3p upregulation causes SIRT1 transcription reduction, thereby causing NF-κB hyperacetylation. As previously mentioned, NF-κB enhances the inflammation response and plays a key role in the onset of microvascular complications associated with DM. The return of SIRT1 to normal concentrations disrupts the cellular metabolic memory caused by high glucose levels through the blockage of NF-κB-mediated pathways [59].

MicroRNA-34a also regulates SIRT1 expression and is implicated in DM development via apoptosis activation and stimulation of inflammation. MiRNA-34a is downregulated by the long non-coding RNA maternally expressed gene 3 (MEG3), thereby causing an increase in SIRT1 levels along with downregulation of NF-κB, TNF-α, IL-6, and IL-1 β pathways which protects cells from damages connected to high glucose. Similarly, MiRNA-377 directly stimulates the NF-κB pathway and inhibits SIRT1. Indeed, in HERCs subjected to high glucose levels, miRNA-377 is correlated with cell migration, thereby facilitating angiogenesis [60].

MicroRNA-211 is another miRNA that inhibits SIRT1, thereby mediating the development of DR through an increase of vascular permeability, RECs apoptosis, and BRB breakdown. Consequently, miRNA-211 may be used as a biomarker to monitor DR onset and represents an interesting potential target for DR therapy [61].

MicroRNA-217 is also linked to DR since it directly downregulates of SIRT1, thus activating the inflammation response in retinal pigment epithelial cells. Inhibition of miRNA-217 was shown to decrease the expression of inflammatory mediators such as TNF-α, IL-6, and IL-1β and suppress NF-κB action in cellular models of diabetes. Notably, these results were shown to be connected to SIRT1 upregulation. SIRT1 overexpression has been recently indicated to possess an antiangiogenic role in DR through the elevation of miRNA-20a levels [62]. MiRNA 20a has been previously indicated to impair branched capillary formation during angiogenesis by inhibiting endothelial cell migration [63]. In addition, overexpression of SIRT1 is correlated with the downregulation of three factors that are pivotal for angiogenesis: YAP, HIF-1α, and VEGFA [64,65,66], hence being crucial for the inhibition of DR development [62,67].

Through the deacetylation of the p65 subunit of NF-κB, SIRT1 impairs the activation of the metalloproteinase-9 (MMP-9). Since DR is characterized by low SIRT1 levels, MMP-9 is activated and promotes oxidative stress and mitochondrial damage, finally leading to capillaries degeneration and pericyte loss [68]. Notably, in mice overexpressing SIRT1, diabetes induction through STZ injection was not associated with vascular and neuronal damages typical of DR. It was rather accompanied by normal vascular density without any sign of vascular leakage [69,70]. Furthermore, retinal cells seemed to be protected from mitochondrial DNA damage induced by high glucose levels and did not display abnormal MMP-9 activation. In STZ-induced diabetic murine models, oxidative stress enhanced the expression of protein arginine methyltransferase 1 (PRMT1) in retinal pigment epithelial cells. Activation of PRMT1 determines SIRT1 downregulation of SIRT1 leading to apoptosis, BRB breakdown, and DR progression [71].

In diabetic patients are systemically augmented the levels of the proinflammatory cytokine IL-17, which stimulates the inflammatory cascade, angiogenesis, and retinal damage and whose expression is modulated by SIRT1. Liu and colleagues showed that higher concentrations of SIRT1 at retinal level were associated with a protective role of SIRT1 in PDR. In peripheral blood mononuclear cells (PBMCs) the authors observed high concentrations of IL-17 combined with a decrease of SIRT1 levels. On the other hand, recent studies indicated that SIRT1 and SIRT3 levels, along with their mRNA concentrations, are increased in PBMCs of diabetic patients [72].

Another regulator of SIRT1 is protein kinase A (PKA) which was shown to upregulate SIRT1 by activating insulin growth factor binding protein 3. In turn, SIRT1 was demonstrated to reduce HMGB1 in HRECs [73].

Endothelin-1 (ET-1) and transforming growth factor-beta (TGF-β) are vasoactive growth factors that play key role in many vascular diseases, including DM. ET-1 isa vasoconstrictive factor and increases vascular permeability, while TGF-β possesses a mitogenic activity and regulates apoptosis and differentiation. Interestingly, SIRT1 decreases ET-1 and TGF-β expression in HRECs exposed to high concentrations of glucose and in STZ-treated murine models [74].

Recently, Zhang et al. reported the protective role ofexendin-4 (EX-4), an analogue of the glucose blood regulator glucagon-like peptide-1 (GLP-1). EX-4 was shown to modulate SIRT1 in rat retinal cells at early diabetes stages. In this animal model, diabetes was associated with elevated ROS levels causing retinal cells death, along with reduced SIRT1 concentrations. Notably, EX-4 administration reduced the formation of H_2_O_2_-induced ROS and increased SIRT1 levels, ultimately leading to a decrease of retinal cell death [75].

A study performed in diabetic rat models by Liu and colleagues demonstrated the beneficial effects in DR of glycyrrhizin (Figure 1), a glycoside extracted from the roots of the licorice plant. Indeed, in rats, this natural product was able to reduce ROS, IL-1β, TNF-α, cleaved caspase 3 levels, along with retinal vasculature permeability. Glycyrrhizin also contributed to the maintenance of normal retinal thickness and cell count. In addition, it could increase the expression of SIRT1 and block the inflammatory cascade triggered by HMBG1 [76].

The naturally occurring polyphenol phytoalexin resveratrol has been shown to increase SIRT1 activity and to possess anticancer, cardioprotective, anti-inflammatory, and antioxidant properties [77].

Resveratrol increases SIRT1 activity by 50% (EC_1.5_) at 46.2 μM, while showing no activity towards SIRT2 and 3 (Figure 1). Resveratrol-mediated SIRT1 activation has been reported to increase life span and augment mitochondrial functions in various mice models [78].

In line with this, previous studies indicated a neuroprotective role of resveratrol since it increases the activity of AMPK. This causes major downregulation of NF-κB, which increases the gene expression of proinflammatory factors responsible for DR pathogenesis [79,80].

Indeed, administration of resveratrol to diabetes murine models considerably reduced retinal leukocyte adhesion induced by diabetes and decreased the retinal expression of VEGF [79,80] and ICAM-1. Furthermore, resveratrol was shown to block glucose-induced apoptosis and the elevation of intracellular ROS levels through activation of the AMPK/SIRT1/PCG-1α (peroxisome proliferator-activated receptor gamma coactivator 1-alpha) pathway [81].

PGC-1α activation leads to an increase of mitochondrial metabolism and the cellular capacity to detoxifying ROS [82,83].

Moreover, resveratrol decreased the acetylation levels of NF-κB p65 subunit, thereby preventing its binding to the MMP-9 promoter, the consequent mitochondrial damage, and the development of DR [84].

It is worth noticing that, given its polyphenolic structure, the beneficial properties of resveratrol may be related to interactions with multiple targets other than SIRT1. The potential of resveratrol in activating SIRT1 and affecting metabolism and oxidative stress, also in the diabetes context, stimulated further research aiming at discovering novel small molecule sirtuin-activating compounds (STACs) [85,86]. These include compounds SRT1460, SRT1720, and SRT2183 (Figure 1), which directly interact with SIRT1 and increase its catalytic activity. Structurally unrelated to resveratrol, these molecules are up to 1000 times more potent in vitro as SIRT1 activators (EC_1.5_ values for SRT1460, SRT1720 and SRT2183 of 2.9, 0.16, and 0.36 µM respectively) and are selective over SIRT2 and 3. Similar to resveratrol, these activators increase cell survival through the stimulation of SIRT1-mediated p53 deacetylation. When tested in diet-induced obese mice, STACs increase glucose tolerance and insulin sensitivity and stimulate mitochondrial capacity [87,88].

Given these activities, STACs have been proposed as potential therapeutic options for the treatment of type 2 diabetes. Overall, the role of SIRT1 in DR has been widely studied and, although a comprehensive knowledge of SIRT1 involvement in DR is still missing [29], the studies conducted so far manage to demonstrate its cardinal role in DR and seem to validate SIRT1 as a promising target for the development of novel therapies (Figure 2).

### 4.2. SIRT3

SIRT3 is mainly located in mitochondria and is involved in metabolic control, energetic homeostasis, and oxidative stress [89].

The expression of manganese superoxide dismutase (MnSOD), one of the most prominent ROS scavenging enzymes, is decreased in DM. Notably, SIRT3 has been shown to activate MnSOD in RECs through direct deacetylation following a reduction of ROS concentration. Similar to SIRT1, SIRT3 has a defensive role against hyperglycemia by inactivating NF-κB-mediated pathways and downregulating the pro-apoptotic protein Bax [89,90].

SIRT3 also activates the FOXO3-mediated anti-free radical pathway [91]. The combined rise in SIRT1 and SIRT3 expression levels is protective towards retinal damages. Moreover, EX-4 administration causes an increase of SIRT3 expression, finally blocking ROS overproduction, and shielding the retinal cells from oxidative damage [76].

Neovascularization is a fundamental step in DR and PDR development. In HRECs, hyperglycemia increases the expression of the neovascularization markers insulin growth factor1, MMP-2, MMP-9, VEGF, and HIF-1α [92,93]. Notably, in the presence of high glucose levels, the induced overexpression of SIRT3 in HRECs diminished the expression of factors that facilitate neovascularization.

In STZ-induced diabetic rats the overexpression of SIRT3 displayed a preventive effect on DR through the downregulation of VEGF expression and the promotion of the expression of autophagy-related factors [94].

Beyond SIRT3, SIRT5 also plays a neuroprotective role in DR. In STZ-induced diabetic mouse models knocked out for SIRT3 and SIRT5, hyperglycemia triggered neuroretinal dysfunction [95]. Nonetheless, the authors did not observe any sign of vascular deterioration which led to the hypothesis that the neuronal dysfunction phase may anticipate the onset of the vascular disease. Interestingly, diabetic mice that were knocked out for either SIRT3 or SIRT5 did not display the same retinal dysfunctions. These results point towards a redundant role of SIRT3 and SIRT5 in retinal neuroprotection [96].

### 4.3. SIRT6

SIRT6 is a nuclear SIRT isoform that interacts with chromatin and finely regulates genome maintenance, response to oxidative stress, inflammation, senescence, and metabolism. Constant exposure to oxidative stress has been demonstrated to lead to the onset of a senescence phenotype [97]. This phenotype consists of a reduction of cell growth and proliferation, along with higher expression of the senescence-associated β-galactosidase. Remarkably, senescence plays a crucial role in the development of DM vascular complications, including DR [98].

Liu and colleagues demonstrated that H_2_O_2_-induced oxidative stress in HRECs determines reduced SIRT6 [58]. In line with this, SIRT6 overexpression led to a reversion of cellular senescence; conversely, SIRT6 knockdown replicated the effects of H_2_O_2_ [99].

In DR, the onset of the vascular disease is preceded by a neurodegenerative phase. This process then causes the breakdown of the BRB which leads to the vascular disease. VEGF is a key protein in the progress of systemic vascular complications of DM and is an important cause of PDR [1,2,3].

Brain-derived neurotrophic factor (BDNF) is a neurotrophic growth factor that modulates neurogenesis and neuronal plasticity. The reduction of BDNF levels in the retina is among the causes of the neurodegenerative step of DR [100]. In diabetic murine models, in early stages of the disease, the expression of both SIRT6 and BDNF was suppressed, whilst retinal VEGF concentrations were augmented, with no signs of vascular disease. This indicates that the absence of neuroprotective factors is among the causes of the neurodegenerative stage. This observation is supported by a prominent reduction of the thickness of the whole retina and of specific retinal layers in mouse models. In addition, VEGF levels were increased in Müller cells in the presence of high glucose concentration and reduced SIRT6 levels. This feedback mechanism suggests that SIRT6 is involved in the development of neurodegenerative symptoms characteristic of early diabetes [101].

Moreover, SIRT6 knockout mouse models showed glycolytic alteration related to an increased apoptosis rate in the inner retinal layers [102].

## 5. Discussion

Vascular inflammation is a pathological condition that affects endothelial cells, pericytes, and immune cells located in the inner layer of blood vessels. During mitochondrial oxidative stress, the endothelial cells of the capillaries react following the activation of an inflammatory process that induces vascular inflammation and dysfunction. In many tissues subjected to stress conditions, macrophages have a central role in inflammation [1,2,3]. In the brain, microglia are made of specialized macrophages able of carrying out phagocytosis to protect CNS neurons. They constitute a network of cells that protects neurons from the surrounding environment. Therefore, neuroinflammation could depend on the continuous activation of glial cells through the action exerted by macrophages. During the activation of the microglial infiltration-specific pro-inflammatory molecules, ROS and toxic molecules are locally released. In type 1 and type 2 diabetes, chronic hyperglycemia leads to the accumulation of advanced end glycation products (AEGP) with consequent endothelial dysfunction and vascular inflammation [103].

Inflammation is quite a defensive response that is triggered by stimuli and harmful conditions, such as infection or tissue damage. In the site where the inflammatory event begins, the involved cells produce a series of cytokines and chemokines that act on the local vascular endothelium, causing dilation of blood vessels, leakage of fluids and recruitment of neutrophils and monocytes from the blood into the tissue. Initial response by resident macrophages produces the release of a variety of inflammatory mediators, including chemokines, cytokines (TNF-α and IL-1β), vasoactive amines, and prostaglandins [1,2,3]. Consequently, local inflammatory exudate is formed, plasma proteins and leukocytes (neutrophils and monocytes) exit the circulation and adhere to the tissues at the site of infection/damage. Once arrived at the damaged tissues, monocytes and neutrophils are activated (either through direct contact with the pathogenic material or through the assistance of cytokines secreted by resident cells) and, to orchestrate a reaction against the onset of the process, release cytotoxic substances (reactive oxygen and nitrogen species, proteases, elastases, collagenases, etc.). These factors, undiscriminating between possible microbial targets and host tissues, can damage the latter as a side effect of the defense activity [104]. Despite it being an essential event for the defense of the integrity of the organism from external attacks, the inflammatory response requires a strict control of its activation but, above all, it must circumscribe the effects of the harmful agent that triggered it, thus avoiding significant damage to the human organism itself [104].

Under stressful conditions, such as hyperglycemia, microglia are activated and reach the neuronal layers and/or the photoreceptor layer [105] triggering the inflammatory process. In retinal tissues, diabetes induces the release of vascular endothelial growth factor (VEGF) which is considered a sign of neovascularization [106,107]. Some studies indicate that the high quality of glycation products in retinal tissues may contribute to the activation of microglia [108]. Gardner reported that during microglia activation, the secretion of VEGF and TNF is increased, contributing to the development of DR [109]. IL-1β is the main cytokine capable of triggering the neuroinflammatory cascade and may have a crucial role in the exacerbation of inflammation. Indeed, IL-1β secretion is abundant in the vascular endothelium as a direct result of chronic hyperglycemia [110]. The latter stimulates endothelial and microglial cells which react not only through activation signals, but also by increasing the production of IL-1β, thus enhancing the inflammatory cascade. In a previous study we have shown that VEGF may be secreted by different types of retinal cells such as EPR cells, pericytes, astrocytes, Müller cells, and endothelial cells. VEGF stimulates the degradation of endothelial cell basement membrane along with their migration with simultaneous release of integrins and MMPs [68].

SIRTs regulate many cellular processes including DNA repair, inflammation, fatty acid oxidation, neurogenesis, carcinogenesis, aging, glucose output, and insulin sensitivity. Their main enzymatic activities involved acetylation, desuccinylation, deproprionylation, demalonylation, demyristoylation, and mono-ADP-ribosylation of both histone and non-histone proteins. The modulation of the activities of these enzymes may open new therapeutic alternatives to treat the pathologies involving SIRTs, including DR. For instance, treatment withmEX-4 increases the expression of SIRT1 and SIRT3, resulting in retinal cell protection from ROS [75,76]. SIRT1 activators, such as the natural product resveratrol and the synthetic STACs, increase mitochondrial function, insulin sensitivity, and cell survival through the stimulation of SIRT1-mediated deacetylation of p53. Moreover, resveratrol enhances AMPK activity, thus controlling the expression of pro-inflammatory factors involved in DR pathogenesis [81].

Microglial cells are activated by intricate interactions between various cell types and pathological pathways. However, the precise mechanism of microglial activation in DR is still elusive. Müller cells play critical roles in extracellular ionic balance, in the glutamate metabolism, and neuronal function [1,2,3]. Low-grade inflammation, activation of immune cells, accumulation of glutamate in the extracellular region, and an altered production of neurotrophic factors orchestrate the onset and progression of retinal neurodegeneration. Neuronal apoptosis and glial activation are the two most significant factors contributing to the development of this disease [1,2,3,13].

In diabetic patients, retinal ganglion cells are susceptible to injury even before the appearance of noticeable microvascular DR lesions [111]. The retinal ganglion cell damage is gradual, with successive severe forms of DR. Patients with diabetes without DR present had reduced GCL and RNFL thickness, indicating that neuroretina alterations occur prior to the manifestation of DR vascular signs [13].

Overall, we described the involvement of SIRTs in DR, which is connected to their various actions in neuronal cells. Increasing evidence indicates that SIRTs are promising therapeutic targets for DR treatment, with their activators that could be employed to stop or at least hold back the degenerative process underlying this disease.

## Figures and Tables

**Figure 1 ijms-23-04048-f001:**
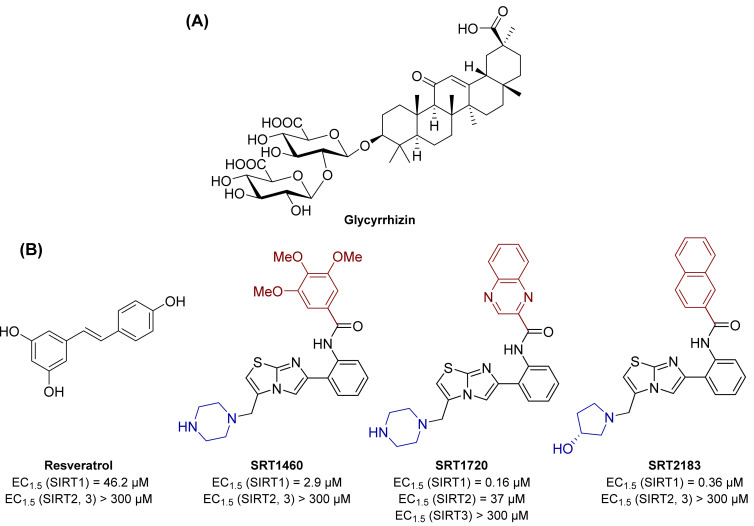
(**A**) Structure of the naturally occurring compound glycyrrhizin, possessing an indirect effect on SIRT1 activity. (**B**) Structures and enzymatic activities of direct SIRT1 activators resveratrol, SRT1460, SRT1720, and SRT2183.

**Figure 2 ijms-23-04048-f002:**
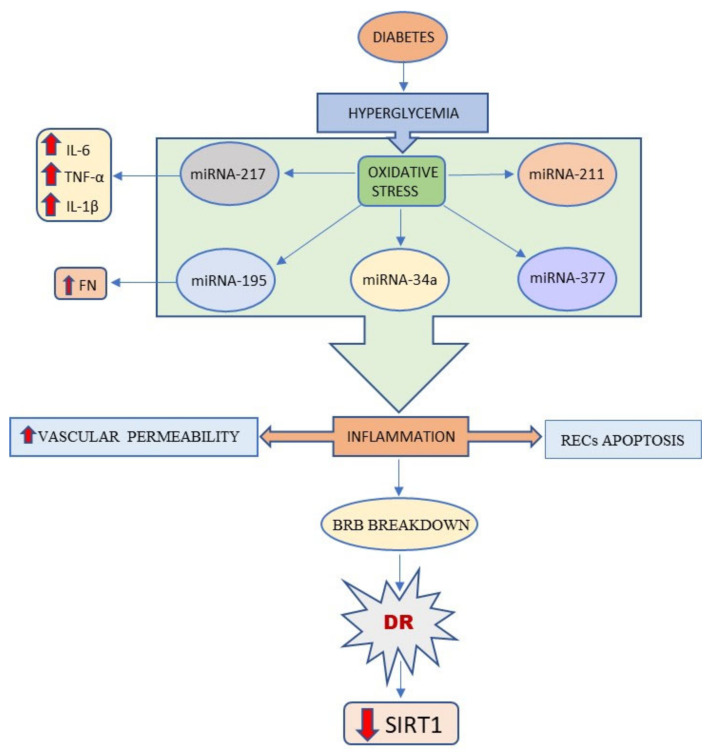
This schema summarizes that Sirt1 expression is reduced in the vascular hyperglycemia.

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
