# Peer review of "Biochemical Functions and Clinical Characterizations of the Sirtuins in Diabetes-Induced Retinal Pathologies"

_ijms, 2022, doi:10.3390/ijms23074048_

Round 1

Reviewer 1 Report

I have finished reviewing the article titled „Biochemical functions and clinical characterizations of the 2 sirtuins in diabetes-induced retinal pathologies” by Samanta Taurone and co-workers. I concluded that the topic is interesting, important and certainly worth publication. I was mostly satisfied with the logic of the text, style and presentation of the data, especially about the supposed function of sirtuins, but I could criticize the description of diabetic retinopathy and the fact that some of the examples listed for their supposed protective role in diabetes are sometimes unclear or even misleading. Overall, there are certain important points that the authors must address before publication could be granted. Belove, I give a detailed description of my points.

First of all – although it is probably the least important – I must comment on the fact that the manuscript was extremely difficult to read. There were certainly at least a few hundred spaces missing from the text making some words fused. I found 2 such case even in the abstract. Due to the mere number, I believe this mistake might have been created during the conversion of the text to PDF format, or caused by some incompatibility, probably on my side. Whatever the cause might be, the text should certainly have to be thoroughly checked before it tis published. I have also noticed some typing errors (eg: line 172: name instead of named) but certainly not all, as they were very difficult to pick, under these circumstances.

I can also criticize the number of abbreviations and their inconsequent usage. There are 72 abbreviations listed at the beginning of the text – that could easily be halved. Some of them are used only once in the text (eg.: SIR, GLP-1, HMGB1 or MEG3) therefore they are completely useless. Some of them used twice (eg.: IBMC) so they might be deleted to make the text easier to digest. Some of the abbreviations that appear at the beginning, I could not find in the text at all – like AICAR and COX-2. Furthermore, not all of them are resolved at the first mention (eg: TNFα – line 163) or not resolved at all (eg: VEGF, PGC-1α). Overall, the complete usage of abbreviations has to be reconsidered throughout the whole text. I believe that this would certainly make to manuscript much easier to follow.

My most important critical comment is about the way diabetic retinopathy (DR) is discussed in the manuscript. It is outdated and not presented in a logical and clear order. First of all, even though the staging and treatment of DR in clinical practice is still mostly based on the detectable vascular alterations, today it is generally accepted, that the neural retina is also compromised in early diabetes parallel to or even preceding vascular pathology (retinal neurodegeneration). Apart from the well-known apoptosis, several retinal cell-types have been shown to be affected morphologically, or functionally – both neurons and glia cells. Besides the tons of evidence derived from several different animal models, functional studies in human patients also show signs indicative of neurodegeneration like color vision defects, ERG abnormalities, increased contrast sensitivity even in patients with early DR or no detectable DR at all (eg.: Wolff et al. 2015; Gella et al. 2015; Harrison et al. 2011) These can precede vasculopathy, or can even be prognostic for the development of DR in the near future. It is unclear at the present, which one of vasculopathy and neuropathy develops in advance of the other, or if they appear parallel, and whether casual relationship exist between them. Some authors even suppose that neural – photoreceptor – damage may be the first step in the pathogenesis of DR and vasculopathy develops only later as a consequence (Tonade et al. 2017; Kern 2017). Irrespectively whichever is the case and what is the authors’ opinion about this, the fact that neurodegeneration exist should certainly be mentioned and discussed. This whole idea should be brought up in the Introduction and not only in the last part of the manuscript, as is the case now. Especially, that the title indicates “diabetes-induced retinal pathologies” which should also involve early, pre-DR phase as well.

This issue is especially important as the protection by sirtuins supposed by the authors may be achieved by affecting also (or even mostly) the neural or glial elements of the retina – and not only the vascular system, as suggested here. Especially, as in some of the models the authors use as reference, neurodegeneration is evident, but only some of the vascular pathology develops.

Furthermore, the description of DR is not too logical, the authors jump from early stage to late proliferative stage in the next sentence, making it difficult for the readers to follow the argument. I also have to mention that the chapter “Clinical manifestation” talks mostly about the pathomechanism and practically anything else but clinics. Either the title or the content should be revised.

I also suggest to clarify some of the statements raised by the authors when discussing the role of surtuins in DR. It is sometimes unclear what type of model was used in the cited literature and to what extent the results are applicable to the retina and DR. A good example illustrating my problem is found in lines 385-386. The cited article is about attention-deficit/hyperactivity disorder and not DR. I am unsure to what extent it demonstrates the statement “Since DR is characterized by low SIRT1 levels, MMP-385 9 is activated and promotes oxidative stress and mitochondrial damage, finally leading to capillaries degeneration and pericyte loss [56]”.

Furter problems with literature:

                - some citation were listed twice (No 20)

                - some listed twice with different numbers (67-68)

In summary, some modifications are clearly needed before publication. I believe that with these corrections incorporated, the manuscript can be significantly improved.

Wolff BE, Bearse MA, Jr, Schneck ME, et al. Color vision and neuroretinal function in diabetes. Doc Ophthalmol. 2015;130(2):131–139.

Gella L, Raman R, Kulothungan V, Pal SS, Ganesan S, Sharma T. Impairment of colour vision in diabetes with no retinopathy: Sankara Nethralaya Diabetic Retinopathy Epidemiology and Molecular Genetics Study (SNDREAMS- II, Report 3). PLoS One. 2015;10(6):e0129391.

Harrison WW, Bearse MA, Jr, Ng JS, et al. Multifocal electroretinograms predict onset of diabetic retinopathy in adult patients with diabetes. Invest Ophthalmol Vis Sci. 2011;52(2):772–777.

Tonade D, Liu H, Palczewski K, Kern TS. Photoreceptor cells produce inflammatory products that contribute to retinal vascular permeability in a mouse model of diabetes. Diabetologia. 2017;60(10):2111–2120.

Kern TS. Do photoreceptor cells cause the development of retinal vascular disease? Vision Res. 2017;139:65–71.

Author Response

Dear Editor and Reviewers,

here enclosed you may find the  REVISED version of our manuscript :ijms-1622357

Title: Biochemical functions and clinical characterizations of the sirtuins in diabetes-induced retinal pathologies

Special Issue: Immune Pathogenesis and Regulation of Ocular Inflammation

Thanks for reviewing our manuscript and for the valuable comments that helped us to clarify some relevant aspects that were missed or unclear in the first version of the paper. We have read the comments of the reviewers and made the changes according to the comments  of the referees. We hope that the corrections performed in the revised manuscript and responses provided below may be helpful to ameliorate this paper adequately.

REVIEWER 1

-First of all – although it is probably the least important – I must comment on the fact that the manuscript was extremely difficult to read. There were certainly at least a few hundred spaces missing from the text making some words fused. I found 2 such case even in the abstract. Due to the mere number, I believe this mistake might have been created during the conversion of the text to PDF format, or caused by some incompatibility, probably on my side. Whatever the cause might be, the text should certainly have to be thoroughly checked before it has been published. I have also noticed some typing errors (eg: line 172: name instead of named) but certainly not all, as they were very difficult to pick, under these circumstances.

RESPONSE:  Thanks for this comment that allowed us to strongly improve the quality of the manuscript.  We have significantly revised the manuscript to remove grammar and typing errors, removing also irrelevant text.

- I have also considered the number of abbreviations and their inconsequent usage. There are 72 abbreviations listed at the beginning of the text – that could easily be halved. Some of them are used only once in the text (eg.: SIR, GLP-1, HMGB1 or MEG3) and therefore they are completely useless. Some of them used twice (eg.: IBMC) so they might be deleted to make the text easier to digest. Some of the abbreviations that appear at the beginning, I could not find in the text at all – like AICAR and COX-2. Furthermore, not all of them are resolved at the first mention (eg: TNFα – line 163) or not resolved at all (eg: VEGF, PGC-1α). Overall, the complete usage of abbreviations has to be reconsidered throughout the whole text. I believe that this would certainly make to manuscript much easier to follow.

RESPONSE:  Thanks for this comment. We have corrected the abbreviations list.

-My most important critical comment is about the way by which  diabetic retinopathy (DR) is discussed in the manuscript. It is outdated and not presented in a logical and clear order. First of all, even though the staging and treatment of DR in clinical practice is still mostly based on the detectable vascular alterations, today it is generally accepted that the neural retina is also compromised in early diabetes in parallel to or even preceding vascular pathology (retinal neurodegeneration). Apart from the well-known apoptosis, several retinal cell-types have been shown to be affected morphologically, or functionally – both neurons and glial  cells. Besides the tons of evidence derived from several different animal models, functional studies in human patients also showed signs indicative of neurodegeneration, like color vision defects, ERG abnormalities, increased contrast sensitivity even in patients with early DR or no detectable DR at all (eg.: Wolff et al. 2015; Gella et al. 2015; Harrison et al. 2011) These can precede vasculopathy, or can even be prognostic for the development of DR in the near future. It is unclear at the present, why one between vasculopathy and neuropathy develops previously than the other, or if they appear parallel, and whether casual relationship exists between them. Some authors even suppose that neural – photoreceptor – damage may be the first step in the pathogenesis of DR and vasculopathy develops only later as a consequence (Tonade et al. 2017; Kern 2017). Irrespectively whichever is the case and what may be the authors’ opinion about this, the fact that neurodegeneration exist should certainly be mentioned and discussed. This whole idea should be brought up in the Introduction and not only in the last part of the manuscript, as is the case considered now. Moreover the title indicates “diabetes-induced retinal pathologies” which should also be involved early, pre-DR phase as well.

This issue is especially important if we consider that  the protection exerted by sirtuins supposed by the authors may be achieved by affecting also (or even mostly) the neural or glial elements of the retina – and not only the vascular system, as suggested here. In particular, as in some of the models the authors have used as reference, neurodegeneration is evident, but only partially the vascular pathology develops.

RESPONSE:Thanks for pointing this out and for suggesting additional informations to be included in our review. We have rewritten the manuscript following this suggestion that helped us clarify some relevant aspects that were missed or unclear in the first version of the paper.

-Furthermore, the description of DR is not too logical, the authors jump from early stage to late proliferative stage in the next sentence, making it difficult for the readers to follow the argument. I also have to mention that the chapter “Clinical manifestation” talks mostly about the pathomechanism and practically anything else but clinics. Either the title or the content should be revised.

RESPONSE:We agree with this comment. We have substantially rewritten the sentence to overcome the issues raised by the reviewer. Thanks for your careful review and comments that helped us significantly improve the quality, flow and content of our review.

-I also suggest to clarify some of the statements raised by the authors when discussing the role of sirtuins in DR. It is sometimes unclear what type of model was used in the cited literature and to what extent the results are applicable to the retina and DR. A good example illustrating my problem is found in lines 385-386. The cited article is about attention-deficit/hyperactivity disorder and not DR. I am unsure to what extent it demonstrates the statement “Since DR is characterized by low SIRT1 levels, MMP-385 9 is activated and promotes oxidative stress and mitochondrial damage, finally leading to capillaries degeneration and pericyte loss [56]”

RESPONSE:Thanks for pointing this out.  Following your suggestion, we  have correct the reference 56.

-Further problems with literature:

                - some citation were listed twice (No 20)

                - some listed twice with different numbers (67-68)

RESPONSE: Thanks for this comment. We have corrected the errors.

Sincerely yours

The Authors

Reviewer 2 Report

This review describes the involvement of SIRTs in DR, outlining their role as potential therapeutic targets in DR. However, some clinical and pathophysiological considerations have to be made:

  1. Lines 123 - 124: not all NPDRs evolve towards PDR. In the current clinical practice, NPDR is classified into 3 categories according to its gravity: mild, moderate and severe and only severe NPDR carries a high risk to transform into PDR if not treated accordingly.
  2. Lines 129 - 130: the growth of new vessels is not subsequent to retinal capillary hyperpermeability, but to capillary occlusions and ischemia.
  3. Lines 148 - 149: DR is classified separately from DME, meaning that DME is not a stage of DR, but it develops independently from DR. Thus, DME can be identified in association to any stage of DR, either NPDR, or PDR.
  4. Line 161: the hypoxic environment is the consequence of capillary occlusions, rather than of the increased permeability of BRB.

Author Response

                                                                                                                   Rome, March23th, 2022

Dear Editor and Reviewers,

here enclosed you may find the  REVISED version of our manuscript :ijms-1622357

Title: Biochemical functions and clinical characterizations of the sirtuins in diabetes-induced retinal pathologies

Special Issue: Immune Pathogenesis and Regulation of Ocular Inflammation

Thanks for reviewing our manuscript and for the valuable comments that helped us to clarify some relevant aspects that were missed or unclear in the first version of the paper. We have read the comments of the reviewers and made the changes according to the comments  of the referees. We hope that the corrections performed in the revised manuscript and responses provided below may be helpful to ameliorate this paper adequately.

REVIEWER2

Lines 123 - 124: not all NPDRs evolve towards PDR. In the current clinical practice, NPDR is classified into 3 categories according to its gravity: mild, moderate and severe and only severe NPDR carries a high risk to transform into PDR if not treated accordingly.

Lines 129 - 130: the growth of new vessels is not subsequent to retinal capillary hyperpermeability, but to capillary occlusions and ischemia.

Lines 148 - 149: DR is classified separately from DME, meaning that DME is not a stage of DR, but it develops independently from DR. Thus, DME can be identified in association to any stage of DR, either NPDR, or PDR.

Line 161: the hypoxic environment is the consequence of capillary occlusions, rather than of the increased permeability of BRB.

RESPONSE:RESPONSE:  Thanks for the thoughtful revision of the manuscript and for all your suggestions that allowed us to strongly improve the quality of the manuscript. We have made all the suggested changes into the text. We believe that after the manuscript revision following your comments and suggestions the quality of this review has substantially improved

The Authors thank the Editor and Reviewers   for the  comments. Their efforts have been  greatly appreciated.

Therefore, in the hope to have clearly and amply answered to all the Editor and Reviewers' comments, which made it possible to significantly improve the quality of the manuscript, the Authors wish that the revised version of the article meets with your approval and  that it will be considered for  publication on IJMS.

We are looking forward to waiting a reply at your earliest convenience.

Sincerely yours

The Authors
